# High niche diversity in Mesozoic pollinating lacewings

Qing Liu[1], Xiumei Lu[2], Qingqing Zhang[1], Jun Chen[1,3], Xiaoting Zheng[3], Weiwei Zhang[4], Xingyue Liu[2] & Bo Wang [1,5]

Niche diversity of pollinating insects plays a vital role in maintaining extant terrestrial ecosystems. A key dimension of pollination niches refers to the insect proboscis length that commonly matches the floral tube length. Here we describe new kalligrammatid lacewings (an iconic Mesozoic pollinating insect lineage) from late Cretaceous Burmese amber and Mesozoic sediments in China. Kalligrammatids display complex configurations of elongate mouthpart elements consisting of well-developed maxillae, labium and their palps. The mouthpart lengths vary among species, from 0.6 to 18.0 mm, suggesting corresponding variability in the floral tube lengths of Mesozoic plants. With the diversification of pollinating habits, the kalligrammatids presented highly divergent traits related to chemical communication and defence mechanisms. Together with other Mesozoic long-proboscid insects, these fossils not only reveal the high niche diversity of Mesozoic pollinating insects but also highlight the diversity of Mesozoic pollinator-dependent plants prior to the rise of angiosperms.

[1] State Key Laboratory of Palaeobiology and Stratigraphy, Center for Excellence in Life and Paleoenvironment, Nanjing Institute of Geology and Palaeontology, Chinese Academy of Sciences, 210008 Nanjing, China. [2] Department of Entomology, China Agricultural University, 100193 Beijing, China. [3] Institute of Geology and Paleontology, Linyi University, 276000 Linyi, China. [4] Three Gorges Entomological Museum, P.O. Box 4680400015 Chongqing, China. [5] Key Laboratory of Zoological Systematics and Evolution, Institute of Zoology, Chinese Academy of Science, 100101 Beijing, China. These authors contributed equally: Qing Liu, Xiumei Lu, Qingqing Zhang. Correspondence and requests for materials should be addressed to X.L. (email: liuxingyue@cau.edu.cn) or to B.W. (email: bowang@nigpas.ac.cn)

Plants and their associated pollinators in extant terrestrial ecosystems display an important ecological interaction in which the pollinators facilitate plant reproduction and thus support the majority of the world's plant diversity[1–4]. Insects are the most important, diverse group of pollinators, and insect pollination played an important role in the evolution of angiosperms via the mechanism of diversity maintenance based on pollination niche partitioning[4–10]. Little is known, however, about ancient pollination insects and their niche diversity during the pre-angiosperm period due to the rarity of fossil evidence of plant–pollinator interactions.

One of the most intensely investigated examples of extant pollination niches is the morphological match between the insect proboscis and floral tube length[5,11]. For example, the distributions of proboscis lengths of hawkmoth assemblages are largely similar to those of floral tube lengths[12]. Therefore, proboscis length can be a fair indicator of pollination niches[5]. Some Mesozoic insects evolved long proboscides, which is a key morphological innovation and provides ideal objects for understanding ancient pollination niches[13–18].

Kalligrammatid lacewings (Neuroptera: Kalligrammatidae) are among the largest and most conspicuous Mesozoic insects[19,20]. They are a group of extinct pollinating insects characterized by their large size, dense crossveins over the entire wing, and distinct eyespots on both the forewings and hind wings, and they are considered to have evolved convergently with modern butterflies in terms of similar wing characters (e.g. wing shape and the presence of eyespots) and siphoning mouthparts[19]. Until now, all previously known kalligrammatids were from two-dimensionally preserved fossils from Western Europe, Central Asia, northeastern China, and Brazil[19–22], to our knowledge, with no species having been described from amber.

Here we report 27 well-preserved kalligrammatid lacewings from late Cretaceous Burmese amber (99 mega-annum [Ma]) and Chinese Early Cretaceous (125 Ma) and Middle Jurassic (165 Ma) compression rocks. Our findings show a more diverse kalligrammatid palaeofauna than previously known, with highly diversified morphological characters, such as bipectinate male antennae, variously prolonged mouthparts and differently developed wing eyespots. Thus these characters provide new insight into the niche diversity, chemical communication and defence mechanisms of these pollinating insects.

## Results
### Systematic palaeontology.

Family Kalligrammatidae Handlirsch
Subfamily Cretanallachiinae Makarkin emend. nov.

**Type genus.** *Cretanallachius* Huang et al.[23]
**Revised diagnosis.** Body small- to medium-sized (forewing length 6–32 mm). Mouthparts siphonate, composed of a short labrum, reduced mandibles and conspicuously elongated maxillae and labium; maxilla with a long blade-like lobe putatively composed of galea or galea+lacinia; maxillary palp much longer, five-segmented, terminal palpomere with blunt tip and ovoid sensory area; labium with elongated ligula; labial palp much longer, three-segmented, terminal palpomere with blunt tip and ovoid sensory area. Male antennae with many bipectinate flagellomeres; female antennae moniliform or filiform. Legs slender; tibiae with tibial spurs, and sometimes with a few spinous setae. Wings broadly subtriangular, with rounded

distal margin; costal space almost entirely broad; trichosors present; nygmata sometimes present on forewings and hind wings; a dark marking, specialized as eyespot in certain species, sometimes present at the middle of wing. Forewing: No branched recurrent humeral veinlet; ScP and RA not fused distad, both distinctly curved posteriad and entering wing margin posteriad pterostigmal area; RP and M fields occupying a broad area, while Cu and A fields relatively narrow; CuA simple or with only a few branches near wing margin; CuP profusely branched, with initial branch proximad its midpoint. Hind wing: Slightly smaller than forewing, in general with similar venation to that of forewing; sigmoid stem of MA sometimes present. Male genitalia: Tergum 9 short, extending ventrad; sternum 9 much longer than tergum 9, distinctly produced posteriad; gonocoxites 9 present as a pair of broad external lobes; ectoprocts paired, with or without callus cerci; gonocoxites 10 present as a pair of slender, internal sclerites, which are distinctly protruding posteriad. Female genitalia: Tergum 9 strongly protruding posteroventrad into a pair of valvate lobes; gonocoxites 9 short and narrow, largely enveloped by valvate lobes of tergum 9, distally with short, digitiform gonostyli 9; ectoprocts paired, with or without callus cerci.
**Genera included.** Five genera: the type genus *Cretanallachius* Huang et al.[23], *Burmogramma* gen. nov., *Burmopsychops* Lu et al.[24], *Cretogramma* gen. nov., and *Oligopsychopsis* Chang et al.[25].

**Remarks.** Cretanallachiinae was originally established and placed in Dilaridae by Makarkin[26], including *Cretanallachius* (as the type genus of the subfamily) and *Burmopsychops*. However, the present finding of the new long-proboscid lacewings (i.e. *Burmogramma* gen. nov. and its relative genera and species) with bipectinate male antennae, posteroventrally protruding female tergum 9 and typical kalligrammatid wing venations as well as eye spots clearly suggests the kalligrammatid affinity of these species that were previously either placed in Psychopsoidea with uncertain familial status[17] or incorrectly placed in Dilaridae[23,26] (See further discussion in Supplementary Note 1).

*Burmogramma liui* gen. et sp. nov.
(Fig. 1 and Supplementary Fig. 1)

**LSID (Life Science Identifier).** urn:lsid:zoobank.org:act: ECF169A1-0941-47A9-88A5-16814C137445; urn:lsid:zoobank.org:act:8E451FD2-FB12-4721-839D-9BC3801C4F8C.
**Type species.** *Burmogramma liui* gen. et sp. nov.
**Material.** Holotype: NIGP164471, a complete male adult. Paratypes: NIGP164472, a nearly complete female adult; NIGP164473, a nearly complete female adult; NIGP164474, a female adult with body partly preserved; NIGP164475, a nearly complete female adult. Earliest Cenomanian amber (99 Ma)[27], Hukawng Valley, northern Myanmar.
**Diagnosis of the genus (and the type species).** Forewing length about 26–32 mm. Pronotum slightly longer than wide. Forewing and hind wing broadly subtriangular with round distal margin, both with a well-developed eyespot medially; trichosors present along distal margin; nygmata absent; forewing costal space with interlink veinlets; RP possessing 17–18 primary branches; MA proximally separating from RP and only with a few short branches distally; MP pectinately branched, occupying a broad subtriangular field, with 5–6 primary branches; anal space relatively broad, A1 densely and pectinately branched; sigmoid stem of hind wing MA absent.

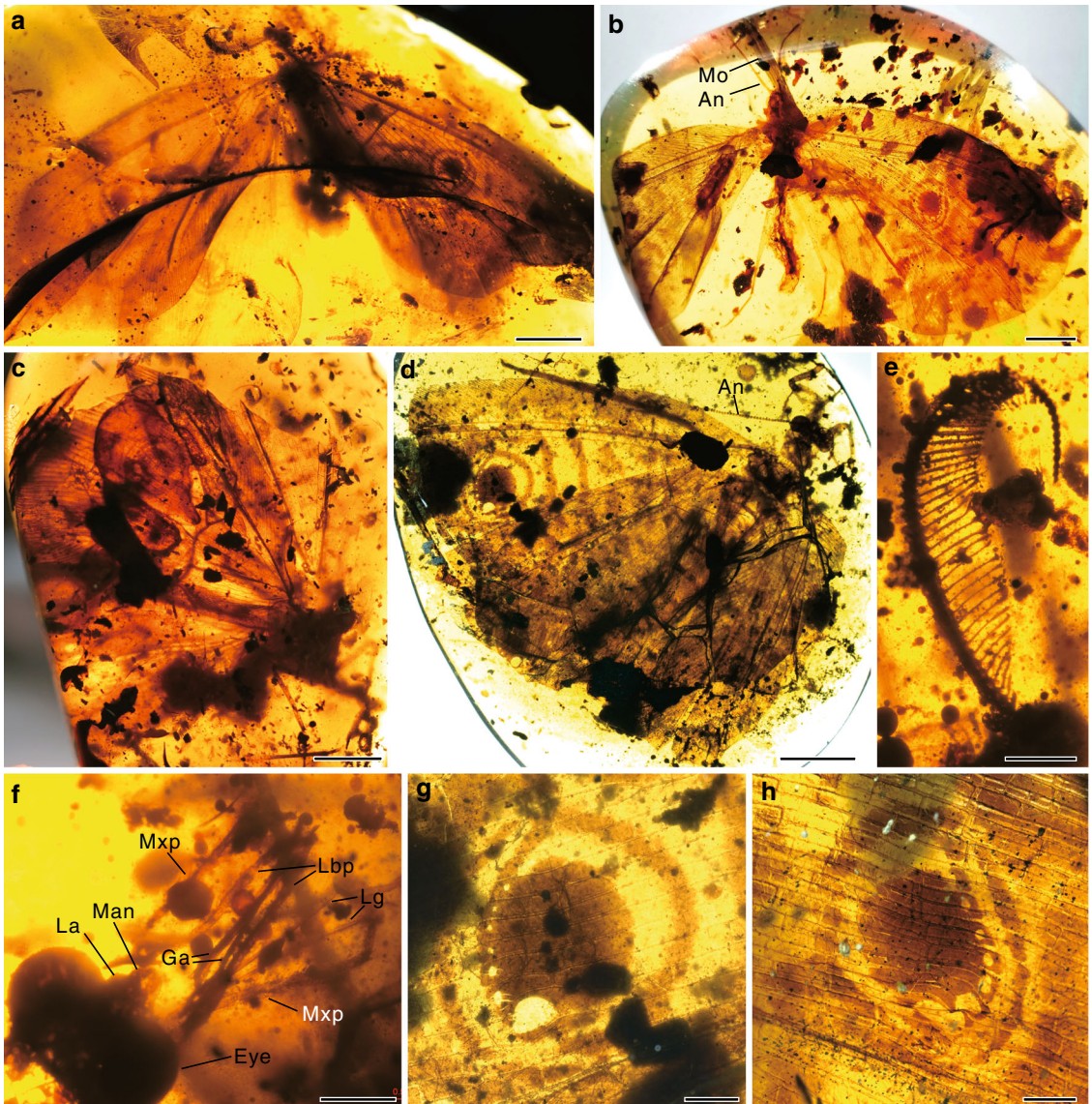

**Fig. 1** *Burmogramma liui* from late Cretaceous Burmese amber. **a**, **e**, **f** Holotype NIGP164471, male, habitus (**a**), antenna (**e**) and mouthparts (**f**).
**b**, **g** Paratype NIGP164472, female, habitus (**b**) and wing eyespot (**g**). **c** Paratype NIGP164473, female. **d**, **h** Paratype NIGP164475, female, habitus (**d**) and wing eyespot (**h**). An antennae, Eye compound eye, Ga galea, La labrum, Lbp labial palp, Lg ligula, Man mandible, Mo mouthparts, Mxp maxillary palp. Scale bars, 5 mm (**a**–**d**), 1 mm (**e**–**h**)

**Etymology.** '*Burmo-*' refers to the occurrence of the type specimen from the Upper Cretaceous of Myanmar (Burma), '*-gramma*' is a traditional suffix of generic names in Kalligrammatidae. The specific epithet '*liui*' is in honour of the collector, Haoying Liu.

**Description.** Holotype male (NIGP164471). Body length 12.00 mm; forewing length 26.0 mm, hind wing length 25.1 mm. Antennae bipectinate. Mouthparts siphonate; labrum short, medially slightly concaved; mandibles absent; maxilla with slenderly elongate, glabrous, blade-like lobe, maxillary palp five-segmented, much longer than galea, palpomere 5 slightly longer than each of other palpomeres; labium with a pair of elongate, thin, glabrous, and distally pointed ligula, labial palp three-segmented, longer than ligula, palpomere 2 nearly equal in length to total length of remaining palpomeres. Pronotum short, slightly wider than long; mesothorax and metathorax robust, mesothorax ~1.6× length of metathorax. Legs slender, with some spinous setae present on tibiae; coxa and trochanter short, femur slightly shorter than tibia; tarsus 5-segmented.

**Forewing.** Broadly subtriangular, with round distal margin. Trichosors present along distal margin. Nygmata absent. Eyespot fuscous and rounded, located distal to wing mid-point, with one large unpigmented spot and several small unpigmented spots arranged along distal margin of eyespot. A nearly circular marking present surrounding eyespot. Costal space broad, ~5.0× width of subcostal space, and slightly narrowed towards wing apex. Costal crossveins inclined and mostly forked near costal margin, with many interlink veinlets among them. Subcostal space slightly narrower than RA space and both regions with about 50 widely spaced crossveins. ScP and RA not fused distad. RP with 18 primary branches; each branch running rectilinearly and with only a few short branches distally. MA separating from RP, with only a few short branches distally. MP pectinately branched from proximal position, with 5–6 primary branches that form a broad, subtriangular field. CuA and CuP diverging proximad initial branching point of MP, nearly parallel with each other on proximal

half, CuA with a few short branches distally, CuP deeply and pectinately branched, with denser branches than CuA. Anal space relatively broad, with at least A1 and A2 densely branched. Crossveins in general numerous, and densely spaced with each other; an outer gradate series of crossveins absent. Hind wing: Broadly subtriangular, with round distal margin; anterodistal portion less produced than that of forewing. Wing venation generally similar to that of forewing. Sigmoid stem of MA undetected, probably absent. Male genital segments not preserved.

Paratypes female (NIGP164472, NIGP164473, NIGP164474, and NIGP164475). Forewing ~32.0 mm long, hind wing slightly shorter than forewing, ~30.5 mm long. Antennae moniliform. Mouthparts siphonate. Pronotum short, slightly wider than long; mesothorax and metathorax robust, mesothorax ~1.6× length of metathorax. Legs slender, with some spinous setae present on tibiae; coxa and trochanter short, femur slightly shorter than tibia; tarsus 5-segmented, with tarsomeres 1–5 gradually shortened; pretarsus with a pair of slender claws and a short arolium bearing paired spinous setae. Wing characters generally same as that in male. Female genital segments: Tergum 9 bilobed dorsoventrally; dorsal part medially separated by a narrow longitudinal suture; ventral part strongly protruding posteriad, forming a pair of valvate lobes that are slightly curved dorsad. Gonocoxites 9 largely invisible, probably enveloped by ventral parts of tergum 9. Ectoprocts paired, proximally largely fused with dorsal part of tergum 9; ovoid callus cerci present, distinctly prominent.

**Remarks.** The new genus undoubtedly belongs to Cretanallachiinae based on the bipectinate male antennae, the siphonate mouthparts, and the female tergum 9 with a pair of large posteroventral lobes. It differs from the other genera of Cretanallachiinae by the relatively large body size, the presence of distinct eyespots on both forewing and hind wing and the broader MP field with 5–6 primary branches.

Genus *Burmopsychops* Lu, Zhang & Liu, 2016
(Fig. 1 and Supplementary Fig. 2)

**Type species.** *Burmopsychops limoae* Lu et al.[24]: 326 (monotypy).
**Revised diagnosis (from Lu et al.[24]).** Forewing length 7–13 mm. Pronotum nearly as long as wide. Forewing and hind wing broadly subtriangular with rounded distal margin, immaculate; nygmata absent; forewing costal space lacking or with few interlink veinlets; RP possessing 7 primary branches; MA proximally separating from R, with only a small marginal fork; MP pectinately branched anteriad, occupying a narrow subtriangular field; anal space narrow; sigmoid stem of hind wing MA absent. Male and female genitalia with callus cerci.

*Burmopsychops labandeirai* sp. nov.
(Fig. 2a and Supplementary Fig. 3)

**LSID (Life Science Identifier).** urn:lsid:zoobank.org: act:2A3B5114-F2D9-4B7B-A3BD-2D032665D44D.
**Materials.** Holotype: NIGP164486, a male adult with distal parts of wings unpreserved. Earliest Cenomanian amber (99 Ma)[27], Hukawng Valley, northern Myanmar.
**Diagnosis.** Forewing length ~13 mm in male. Forewing and hind wing broadly subtriangular, immaculate; nygmata absent; a few interlink veinlets present on distal half of costal space of both forewing and hind wings. Male gonocoxites 9 without internal teeth; gonocoxites 10 paired, anterior half distinctly

directed anterolaterally, posterior half directing posterodorsad, with serrate apices.
**Etymology.** The new species is dedicated to Dr. Conrad C. Labandeira for his great contribution on the early evolution of plant–insect associations.

**Description.** Holotype male (NIGP164486). Body length 9.0 mm; forewing with preserved part 12.0 mm long, entirely ~13.0 mm by estimation. Antennae bipectinate. Mouthparts siphonate. Pronotum short; mesothorax and metathorax robust, mesothorax ~2.0× length of metathorax. Legs slender, with some spinous setae present on tibiae; tarsus five-segmented, with tarsomeres 1–5 gradually shortened; pretarsus with a pair of slender claws and a short arolium bearing paired spinous setae.

**Forewing**. Incomplete. Broadly subtriangular by estimation. Transparent and immaculate. Nygma absent. Costal space broad, ~6.0× width of subcostal space, with some crossveins on distal half forked and a few interlink veinlets; a simple humeral veinlet slightly bent to wing base. Subcostal space slightly narrower than RA space, with 12 widely spaced crossveins preserved. RA space with 12 widely spaced crossveins preserved. RP with at least 9 primary branches. MA separating from R, initially branched proximad its midpoint. MP pectinately branched from proximal position, with five main branches that form a narrow, subtriangular field. CuA and CuP diverging proximad initial branching point of MP, nearly parallel with each other on proximal half, CuA with two short, marginally forked, branches distally, CuP deeply and pectinately branched, with at least four branches. Anal space short and narrow. Hind wing: Incomplete. Transparent and immaculate. Venation in general similar to that of forewing. Sigmoid stem of MA absent.

**Male genital segments**. Tergum 9 short, arched, distinctly extending ventrad, dorsally separated by a medial longitudinal incision. Sternum 9 much longer than tergum 9, subtriangular, posteromedially with a clavate projection. Gonocoxites 9 paired, broadly valvate, distinctly narrowed posteriad, with blunt tips, but lacking internal teeth. Ectoprocts paired, flatly subtriangular, amalgamated with tergum 9; semiglobular calluls cerci present on posterior margin of ectoprocts. A pair of strongly sclerotized sclerites (putative gonocoxites 10) present internally, largely enveloped by gonocoxites 9; each of them with anterior half extending anterolaterally and with posterior half nearly vertical to anterior half, protruding posterodorsad, with a few minute dents at tip. An indistinct, slender, transversely arched sclerite (putative fused gonocoxites 11) present internally, with lateral parts seeming touching gonocoxites 9.

**Remarks.** The new species differs from *B. limoae* by the larger body size. In male of the new species, the forewing length is ~13 mm, while the female forewing length (usually longer than that of conspecific males) in *B. limoae* only ranges 7.7–10.0 mm.

Genus *Cretanallachius* Huang et al., 2015.
(Fig. 2d and Supplementary Figs. 4 and 5)

**Type species.** *Cretanallachius magnificus* Huang et al.[23]: 275 (monotypy).
**Revised diagnosis (from Huang et al.[23]).** Forewing length ~6–8 mm. Pronotum slightly wider than long. Forewing and hind wing broadly subtriangular with rounded distal margin, immaculate; forewing costal space with most crossveins simple or only forked near costal margin, interlink veinlets absent; RP possessing 5–6 primary branches; MA proximally separating from RP, initially branched nearly at or beyond its midpoint; MP with 2 primary branches, which are parallel with each other for a

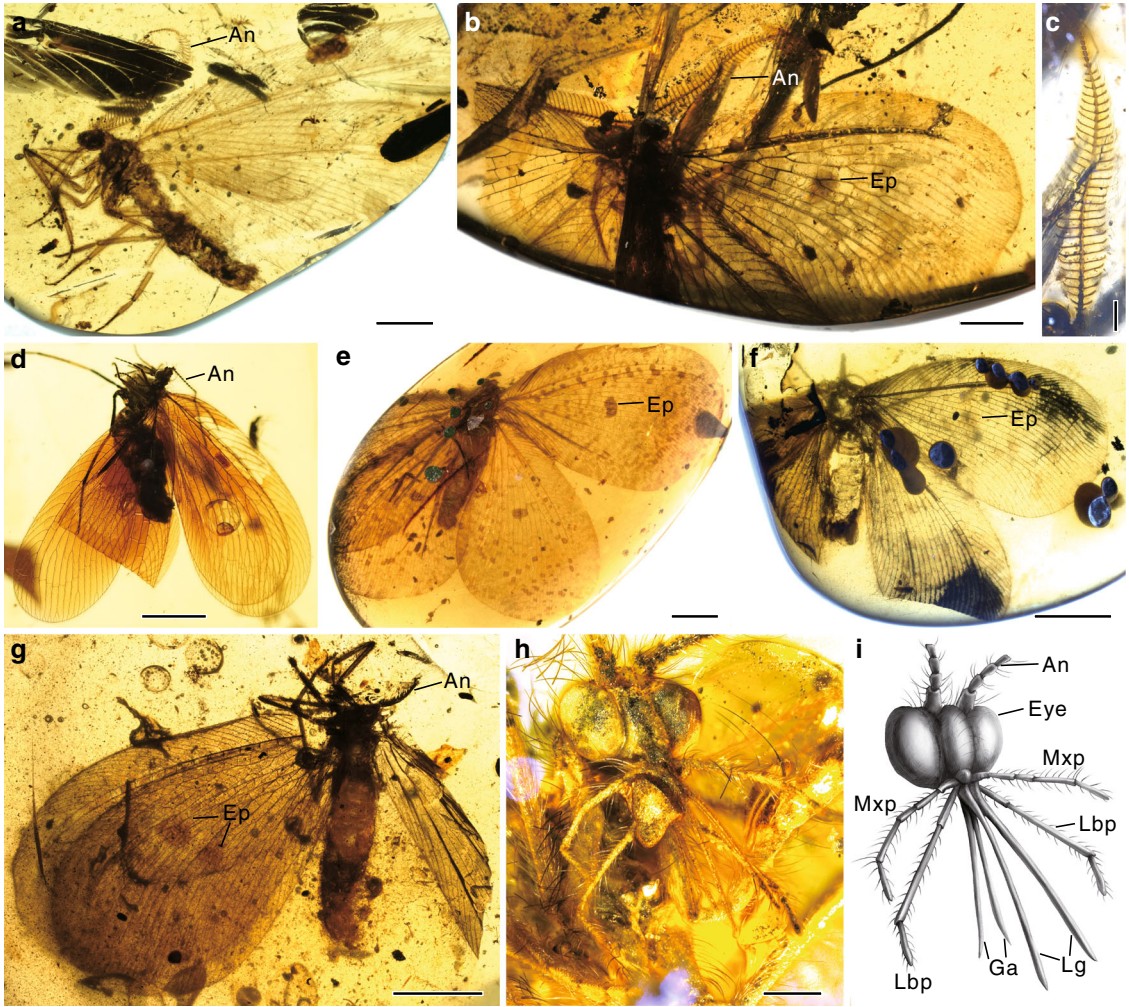

**Fig. 2** Cretanallachiinae from late Cretaceous Burmese amber. **a** *Burmopsychops labandeirai*, holotype NIGP164486, male. **b**, **c** *Cretogramma engeli*, holotype NIGP164481, male, habitus (**b**) and antennae (**c**). **d** *Cretanallachius magnificus*, CAU-BA-XF-18003, female. **e** O*ligopsychopsis grandis*, holotype, CAM BA-0010, female. **f** *Cretogramma engeli*, paratype NIGP164483, female. **g** *Oligopsychopsis groehni*, NIGP164479, female. **h**, **i** O*ligopsychopsis groehni*, CAU-BA-WN-18001, photo and drawing of mouthparts. An antennae, Eye compound eye, Ep eyespot, Ga galea, Lbp labial palp, Lg ligula, Mxp maxillary palp. Both **h** and **i** to scale. Scale bars, 2 mm (**a**, **b**, **d**), 4 mm (**e**–**g**) and 0.5 mm (**c**, **h**)

long distance; anal space narrow; sigmoid stem of hind wing MA present. Callus cerci present in male but absent in female.

*Cretogramma engeli* gen. et sp. nov.
(Fig. 2b, c, f and Supplementary Fig. 6)

**LSID (Life Science Identifier).** urn:lsid:zoobank.org:act: EB82C6CB-D531-40AB-BBD5-4AE0501A353C; urn:lsid: zoobank.org:act:12666256-DCB1-4DC1-85F0-510E771CB372.

**Type species.** *Cretogramma engeli* gen. et sp. nov.

**Materials.** Holotype: NIGP164481, a nearly complete male adult. Paratypes: NIGP164482, a nearly complete male adult. NIGP164483, a female adult; NIGP164484, a female adult. Earliest Cenomanian amber (99 Ma)[27], Hukawng Valley, northern Myanmar.

**Diagnosis of the genus (and the type species).** Forewing length ~12–14 mm. Pronotum slightly wider than long. Forewing and hind wing broadly subtriangular with round distal margin, both medially with an ovoid dark spot; a nygma present medially at least on forewing; forewing costal space with interlink veinlets; RP possessing 6–10 primary branches; MA proximally separating from R, and initially branched distad its midpoint, with only two primary branches; MP pectinately branched anteriad, occupying a narrow subtriangular field; anal space narrow; sigmoid stem of hind wing MA absent.

**Etymology.** From 'creto-' (Cretaceous) and '-gramma' (a traditional suffix of generic names in Kalligrammatidae) in reference to the geological period of the new genus. The specific epithet 'engeli' is in honour of Dr. Michael S. Engel for his great contribution on fossil insects.

**Description.** Holotype male (NIGP164481). Body length ~8.0 mm; forewing length 11.6 mm. Antennae bipectinate. Mouthparts siphonate; labrum short, medially slightly concaved; mandibles absent; maxilla with slenderly elongate, glabrous, blade-like lobe, maxillary palp five-segmented, much longer than galea, palpomere 5 slightly longer than each of other palpomeres; labium with a pair of elongate, thin, glabrous, and distally pointed ligula, labial palp three-segmented, longer than ligula,

palpomere 2 nearly equal in length to total length of remaining palpomeres. Pronotum short, slightly wider than long; mesothorax and metathorax robust, mesothorax ~2.0× length of metathorax. Legs slender, with some spinous setae present on tibiae; coxa and trochanter short, femur slightly shorter than tibia; tarsus five-segmented, with tarsomeres 1–5 gradually shortened; pretarsus with a pair of slender claws and a short arolium bearing paired spinous setae.

**Forewing**. Broadly subtriangular, with round distal margin. An ovoid dark marking present medially, with obscure margin. Trichosors present along distal margin. A nygma present at middle of wing. Costal space broad, ~5.0× width of subcostal space, with most crossveins on distal half deeply forked and a few interlink veinlets; a simple humeral veinlet slightly bent to wing base. Subcostal space slightly broader than RA space, with 13 widely spaced crossveins. RA space with nine widely spaced crossveins. ScP and RA not fused distad. RP with six primary branches; posterior three branches deeply forked. MA separating from R, initially branched distad its midpoint, with only two primary branches, each of which is bifurcated distad. MP pectinately branched from proximal position, with three main branches that form a narrow, subtriangular field. CuA and CuP diverging proximad initial branching point of MP, nearly parallel with each other on proximal half, CuA with two short, marginally forked, branches distally, CuP deeply and pectinately branched, with at least four branches. Anal space short and narrow. Crossveins in general numerous and widely spaced with each other; an outer gradate series of crossveins present. Hind wing: Largely not preserved. Male genital segments: Not preserved.

Paratype male (NIGP164482). Forewing length ~12.0 mm. Body poorly preserved, with male genitalia partly preserved. Male genital segments: Gonocoxites 9 broadly valvate, distally with some stout setae. Gonocoxites 10 slender, distally forked into a slender, spinous projection and a robust, terminally bifurcated projection.

Paratype females (NIGP164483 and NIGP164484). Forewing broad and ovate, ~14.0 mm in length. Eyespot obscure and nearly rounded, located slightly basal to wing mid-length. Crossveins present over entire wing. Costal space broad and narrowed towards wing apex. Costal veinlets sinuous with several basal branches simple and all others forked. Subcostal space slightly narrower than RA space, and both regions with widely spaced crossveins. ScP and RA not fused distally. RP with ten primary branches; each branch running rectilinearly and forked distally several times except for fifth RP branch forked much basally. MA forked distal to wing midpoint. MP well developed and forked basal to MA. CuA simple and forked distally. CuP forked much more basal to CuA and dichotomously forked several times. A1 forked slightly proximad its mid-length. A2 well developed. A3 short.

**Remarks.** See remarks of *Oligopsychopsis* for morphological comparison between the new genus and its closely related genus.

Genus *Oligopsychopsis* Chang et al., 2018
(Fig. 2e, g–i and Supplementary Figs. 7 and 8)

**Type species.** *Oligopsychopsis penniformis* Chang et al.[25]: 534 (monotypy).
**Other species.** *Oligopsychopsis grandi* sp. nov.; *Oligopsychopsis groehni* (Makarkin[26]), comb. nov.
**Revised diagnosis (from Chang et al.[25]).** Forewing length ~10–20 mm. Pronotum slightly wider than long. Forewing and hind wing broadly subtriangular with rounded distal margin, both medially with an ovoid dark spot (probably being reduced condition of the typical kalligrammatid eyespot); a nygma present medially on both forewing and hind wing; forewing costal space with interlink veinlets; RP possessing 6–8 primary branches; MA proximally separating from R and deeply branched with 3–4 primary branches; MP pectinately branched anteriad, occupying a narrow subtriangular field; anal space narrow; sigmoid stem of hind wing MA present at least in some species. Callus cerci present at least in female.

*Oligopsychopsis grandis* sp. nov.
(Fig. 2e and Supplementary Fig. 8)

**LSID (Life Science Id entifier).** urn:lsid:zoobank.org:act: A8763301-9841-4F69-9F77-8022D3564C77.
**Materials.** Holotype: CAM BA-0010, a complete female adult. Paratypes: CAM BA-0011, a nearly complete male adult; NIGP164480, female. Earliest Cenomanian amber (99 Ma)[27], Hukawng Valley, northern Myanmar.
**Diagnosis.** Forewing length ~15 mm in male and ~20 mm in female. Forewing and hind wing broadly subtriangular with rounded distal margin, both medially with an ovoid dark spot and with many distinct dark spots distad median ovoid marking; a nygma present medially on both forewing and hind wing; forewing costal space with interlink veinlets; RP possessing 6–8 primary branches; MA proximally separating from R and deeply branched with 3–4 primary branches; MP pectinately branched anteriad, occupying a narrow subtriangular field; anal space narrow; sigmoid stem of hind wing MA present.
**Etymology.** The specific epithet 'grandis' refers to the relatively larger body size of the new species.

**Description.** Holotype female (CAM BA-0010). Body length 12.0 mm; forewing length 20.0 mm, hind wing length 19.0 mm. Antennae moniliform. Mouthparts siphonate. Pronotum short, slightly wider than long; mesothorax and metathorax robust, mesothorax ~2.0× length of metathorax. Legs slender, some spinous setae present on tibiae.

**Forewing**. Broadly subtriangular, with round distal margin. An ovoid dark marking present medially, and many smaller dark spots present along subcostal and RA spaces as well as on distal half of wing. Trichosors present along distal margin. A nygma present at the middle of wing. Costal space broad, ~5.0× width of subcostal space, with most crossveins deeply forked and with many interlink veinlets among them; a simple humeral veinlet slightly bent to wing base. Subcostal space slightly broader than RA space, with 18 widely spaced crossveins. ScP and RA not fused distad. RA space with 12 widely spaced crossveins. RP with eight primary branches; posterior three branches deeply forked. MA separating from R, deeply and dichotomously branched. MP pectinately branched from proximal position, with four primary branches that form a narrow, subtriangular field. CuA and CuP diverging proximad initial branching point of MP, nearly parallel with each other on proximal half, CuA with three short branches distally, CuP deeply and pectinately branched, with much denser branches than CuA. Anal space short and narrow; A1 deeply and dichotomously branched into four branches; A2 deeply and pectinately branched into four branches; A3 trifurcate. Crossveins in general numerous and widely spaced with each other; an outer gradate series of crossveins present. Hind wing: Broadly subtriangular, with rounded distal margin; proximal portion strongly narrowed. Wing venation and marking pattern generally similar to that of forewing. Sigmoid stem of MA present.

**Female genital segments**. Tergum 9 bilobed dorsoventrally; dorsal part invisible; ventral part strongly protruding posteriad, forming a pair of valvate lobes that are slightly curved dorsad and nearly twice as long as wide. A narrow sclerite (putative gonapophysis 8) that is acutely tapering posteriad present between bases of ventral parts of tergum 9. Gonocoxites 9 largely

invisible, enveloped by ventral parts of tergum 9. Ectoprocts with only lateral portion of left ectoproct discernible; an ovoid callus cercus present.

Paratype male (CAM BA-0011). Body length 9.0 mm; forewing length 15.0 mm, hind wing length 13.0 mm. Non-genital morphology generally same as that in female except for bipectinate antennae. Male genital segments: Very incompletely preserved. Tergum 9 slightly shorter than tergum 8, dorsally divided by a longitudinal incision. Partially preserved putative ectoproct broad, dorsally with a row of teeth.

**Remarks.** The new species differs from *O. groehni* by the relatively large body size, the more distinct wing markings (especially those on distal part of wing) and the relatively broad ventral lobes of female tergum 9. However, the new species can be distinguished from *O. penniformis* by the relatively small body size and the distinct wing marking patterns.

Family Incertae sedis
Genus *Fiaponeura* Lu, Zhang & Liu, 2016
(Supplementary Fig. 9)

**Type species.** *Fiaponeura penghiani* Lu et al.[24]: 325 (monotypy).
**Revised diagnosis (from Lu et al.[24]).** Forewing length ~12 mm. Antenna filiform in both male and female. Mouthparts composed of a bifid labrum, reduced mandibles and conspicuously elongated maxillae and labium; maxilla with a long blade-like lobe putatively composed of galea or galea +lacinia, ~2/3 the length of maxillary palp; labium with elongated, distally bifid ligula, which is ~2/3 the length of labial palp. Pronotum about twice as long as wide. Forewing broadly subtriangular with round distal margin and with transversely band-like dark markings; trichosors present along almost entire wing margin; a proximal nygma present; costal space with many forked crossveins, interlink veinlets among them absent; ScP and RA fused distally; 4–5 oblique radial branches (ORBs) and MA diverging from R; MA dichotomously branched from its midpoint; MP deeply and dichotomously branched, occupying a narrow subtriangular field. Hind wing with transversely band-like dark markings; a nygma present at proximal 1/3; ScP and RA fused distally; sigmoid stem of MA present.

**Phylogenetic analysis.** The parsimony analysis of the primary matrix (matrix 1) yielded eight most parsimonious trees (MPT) (Fig. 3 and Supplementary Figs. 11 and 12; length = 99, consistency index = 40, retention index = 72). All genera of Psychopsoidea are assigned to a monophyletic group. Being different from the result in Lu et al.[17], all Burmese amber long-proboscid psychopsoids, except *Fiaponeura*, together with the genera of Kalligrammatidae and Aetheogrammatidae, form a monophyletic clade. However, the relationships among *Fiaponeura*, Osmylopsychopidae, Psychopsidae and the above clade are poorly resolved. Among the

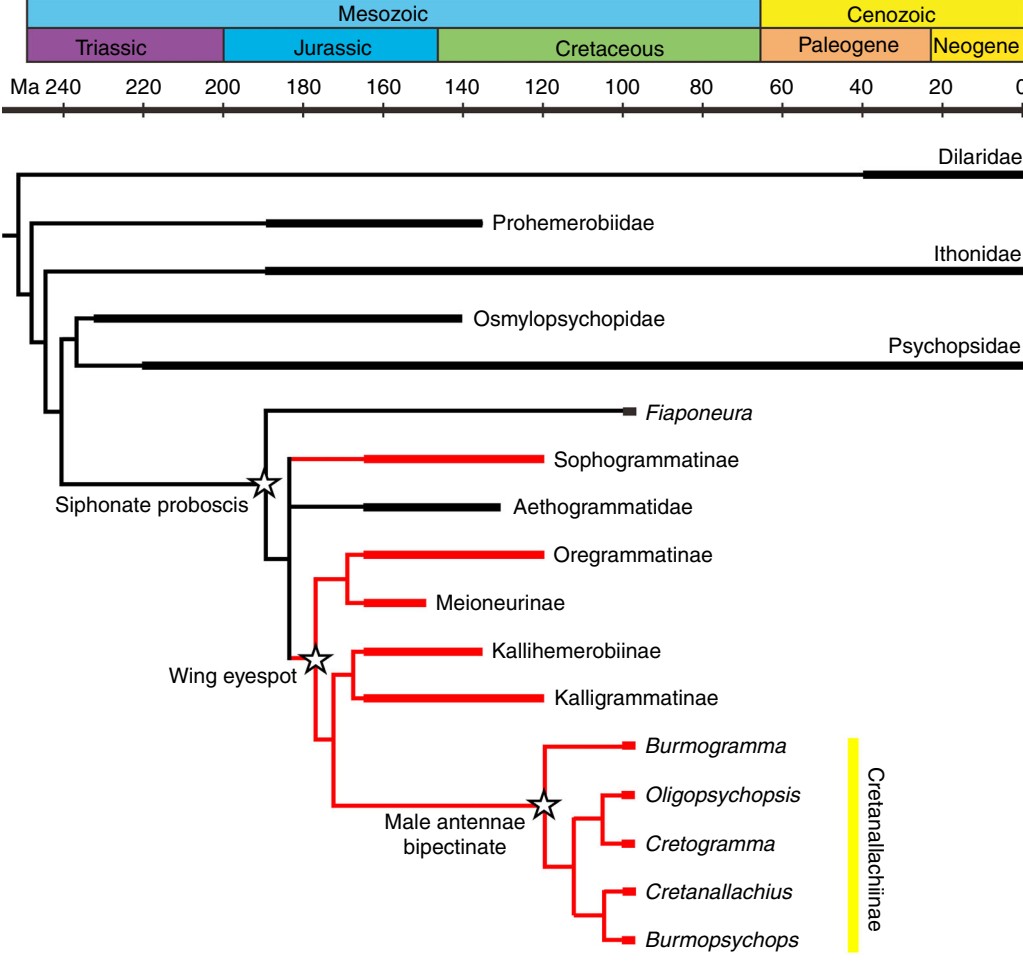

**Fig. 3** Evolutionary history of kalligrammatids and related groups. Thick lines indicate the known extent of the fossil record. Branches representing kalligrammatids are red. See Supplementary Information for details of the phylogenetic analysis

eight MPTs, six of them recover the sister group relationship of Osmylopsychopidae and Psychopsidae, and the sister group relationship of *Fiaponeura* and the clade comprising Kalligrammatidae, Aetheogrammatidae, and most Burmese amber long-proboscid psychopsoids. The remaining two MPTs recover the sister group relationship between Osmylopsychopidae and the clade comprising Kalligrammatidae, Aetheogrammatidae and most Burmese amber long-proboscid psychopsoids, which is less likely considering the obvious close relationship between Osmylopsychopidae and Psychopsidae (see Lu et al.[17]). Furthermore, considering the close relationships between *Electropsychops* and *Litopsychopsis* (see Lu et al.[17]) and between *Meioneurites* and *Oregramma* (see Yang et al.[22]), only one MPT stands as the preferable tree for the following discussion.

The Burmese amber long-proboscid psychopsoids, except *Fiaponeura*, form a monophyletic group that was assigned to be the sister group of *Kallihermerobius*+*Kalligramma*. *Burmogramma* gen. nov. was assigned to be the sister group of the remaining four Burmese amber long-proboscid genera, in which *Burmopsychops* is the sister group of *Cretanallachius*, while *Oligopsychopsis* is the sister group of *Cretogramma* gen. nov. It is notable that the autapomorphies of this monophyletic group do not include the bipectinate male antennae and the specialized female tergum 9, probably due to the missing data in the other kalligrammatid taxa. Nevertheless, these two character states are recovered to be among the autapomorphies of the above monophyletic group in the parsimony analysis of matrix 2 (Supplementary Figs. 13 and 14; eight MPTs generated with the same topologies as those from the analysis of matrix 1, length = 99, consistency index = 40, retention index = 72).

The Burmese amber long-proboscid lacewing species were previously either placed in Psychopsoidea with uncertain familial status[17,25] or placed in Dilaridae[23,27]. The result of our phylogenetic analysis suggests the kalligrammatid affinity of Cretanallachiinae based on the reduction of mandibles (char. 24:1), the subtriangular branching field of forewing vein MP (char. 11:1) and the presence of eyespots (char. 17:1), although in some small-sized Cretanallachiinae species the forewing MP branches are reduced and the eyespots are feebly developed or even lost. However, the familial placement of *Fiaponeura* from the Burmese amber remains uncertain, although it is probably

close to Kalligrammatidae and Aetheogrammatidae but does not belong to Cretanallachiinae based on our phylogenetic result.

## Discussion

The earliest kalligrammatids are known from the Jurassic Daohugou beds (Callovian–Oxfordian) of China (Supplementary Fig. 10)[22,28], and the latest species is from the Lower Cretaceous Crato Formation (Aptian–Albian) of Brazil[29]. Earlier studies suggested that kalligrammatids became extinct during the Aptian–Albian gap (125–100 Ma)[19,30], but our amber fossils provide the latest occurrence of kalligrammatids (Fig. 3) and further demonstrate that this group was diverse in the late Cretaceous.

The kalligrammatid proboscis was thought to consist of maxillary galeae conjoined to form a tubular siphon anatomically similar to that of the lepidopteran Glossata[19], but the two-dimensional preservation of mouthparts in compression fossils commonly hinders a detailed anatomical investigation. Instead, microscopic preservation in amber shows that the proboscis is much more complex than previously thought. The proboscis in Cretanallachiinae has two pairs of long galeae (or galeae+laciniae) and ligulae, and along the entire length of the galea is a series of at least 40 thin, transverse, cuticular bands separated by thinner bands of membrane (Fig. 2h, i and Supplementary Fig. 2b). This structure is highly convergent with the galeae of the long-tongued Cretaceous scorpionfly, genus *Parapolycentropus*[31], and it demonstrates that the galeae could temporarily come together and enclose the ligulae, thus forming a functional proboscis (e.g. Fig. 1b, f). The cuticular bands allow the proboscis to curve significantly but not coil as in Lepidoptera[32], because there is no intrinsic or extrinsic musculature in the galeae. The pair of ligulae can also be locked together to form a food canal[17]. The galeae are slightly longer than the ligulae; therefore, for the ligulae to reach food, the terminal parts of the galeal valves would sometimes need to separate.

Only one pair of maxillary palps was previously reported in kalligrammatids[19,22], but our investigation shows that both maxillary and labial palps are well developed in Cretanallachiinae and distinctly longer than the proboscis (Supplementary Table 1). These palps are densely covered with long setae, and there is an ovoid sensory area in the terminal palpomeres (Fig. 2h, i and Supplementary Fig. 2b). Such features suggest that these palps

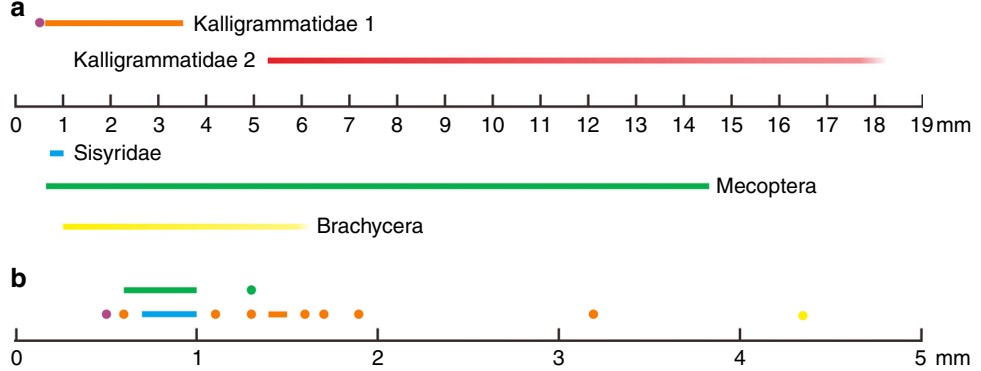

**Fig. 4** Proboscid length of Mesozoic long-proboscid insects. **a** All known Mesozoic long-proboscid insects. The purple dot represents *Fiaponeura*. The orange bar (Kalligrammatidae 1) represents the range of proboscid lengths of late Cretaceous kalligrammatids from Burmese amber. The red bar (Kalligrammatidae 2) represents the range of proboscid lengths of Jurassic and Early Cretaceous kalligrammatids from rocks. The blue bar (Sisyridae) represents the range of proboscid lengths of Cretaceous spongillaflies. The green bar (Mecoptera) represents the range of proboscid length of Jurassic and Cretaceous long-proboscid scorpionflies. The yellow bar (Brachycera) shows the range of proboscid lengths of Jurassic and Cretaceous long-proboscid brachyceran flies. **b** Long-proboscid insects in late Cretaceous Burmese amber. The purple dot represents *Fiaponeura*. Orange and yellow dots represent kalligrammatids and flies, respectively, and blue and green bars represent spongillaflies and scorpionflies, respectively. See Supplementary Tables 1 and 2 for details

were probably used to probe for nectar or pollen[33,34]. Furthermore, these palps are also present in three well-preserved kalligrammatid specimens from the Jurassic and Cretaceous of China (Supplementary Fig. 10) as well as in a specimen from Central Asia[20], suggesting that both maxillary and labial palps were present in most or even all kalligrammatids.

Kalligrammatids represent a distinctive pollination strategy involving gymnosperms[19]. The lengths of kalligrammatid proboscides vary greatly, from 0.6 to 3.2 mm in amber inclusions and from approximately 5 to 18 mm in compression fossils (Fig. 4). Insect proboscis length can account for the floral tube length[12]. The high diversity of kalligrammatids and large variation in their proboscis lengths strongly suggest diverse plant hosts with different floral tube lengths. In the ecologically analogous Mesozoic scorpionflies and brachyceran flies, the lengths of their proboscides also varied distinctly, ranging from approximately 0.6 to 14.5 mm and from 1 to 6 mm, respectively (Fig. 4a; Supplementary Table 2). Interestingly, for long-proboscid insects in Burmese amber, proboscis lengths vary among most species (Fig. 4b), which most likely corresponds to different host plants and diverse pollination niches. Therefore, niche diversity was high in Mesozoic pollinating insects prior to the rise of angiosperms. This high niche diversity suggests that pollination niche partitioning may have been present among some Mesozoic insects. If pollination niches were partitioned, as in extant ecosystems[11], this likely increased pollination effectiveness and reduced costs in pollination mutualisms, thus contributing to the high diversity of pollinating insects and the success of pollinator-dependent plants during the Cretaceous period.

In Kalligrammatidae, the Burmese amber species (Cretanallachiinae) represent the latest record in kalligrammatid evolutionary history and a relatively advanced lineage within the family. In addition to the diversified feeding habits associated with different host plants, some other adaptive traits also played a significant role in the survival of Burmese amber kalligrammatids during the mid-Cretaceous when the angiosperms were diversifying.

Remarkably, male Cretanallachiinae have the ramified antennae, which are common (including pectinate, bipectinate, plumose and flabellate) in living insects, but are extremely rare in Mesozoic insects, with a few examples known only in males[35–37]. Our findings confirm that Cretanallachiinae display sexually dimorphic antennae that are bipectinate in males but moniliform or filiform in females (Figs. 1 and 2). In Neuroptera, similar sexually dimorphic antennae are only known in Dilaridae, and the pectinate male antennae support the monophyly of Dilaridae[38]. The male antennae are unipectinate in Dilaridae but bipectinate in Cretanallachiinae. According to the distantly related phylogenetic relationships between these two groups, the sexually dimorphic antennae had evolved independently in Dilaridae and Cretanallachiinae, representing evolutionary convergence.

Bipectinate antennae represent an overall expansion in the antennal surface area associated with an increase in the number of olfactory sensillae that help males identify sex pheromones released as attractants from females[39]. The sexually dimorphic antennae in various heterogeneous insect groups are a well-known example of convergent evolution in response to pre-mating chemical communication[40]. The Cretanallachiinae species, together with the contemporary dilarid species from the Burmese amber[38], present definitive evidence of sexually dimorphic antennae and demonstrate the antiquity of long-distance mate-searching behaviour in insects.

The wing eyespot (eye-like marking) is another specialized character that is widespread in kalligrammatids (Figs. 1g, h and 2b, d–g)[19], whereas it is very rarely developed in extant Neuroptera[41,42]. The primary function of eyespots is predation

deterrence by producing an effective startle response to ward off potential predators[43]. By the late Cretaceous, many new predaceous arthropods[44,45] (including some spiders, dragonflies and ants) and vertebrates[46] (including some lizards, birds and mammals) had evolved. Wing eyespots were an important defensive mechanism in kalligrammatids. However, eyespots are mostly restricted to large kalligrammatids (forewing length >25 mm) (e.g. Fig. 1g, h), and they are weak in small species of Cretanallachiinae, being almost invisible in the smallest species (e.g. Figs. 1g, h and 2b, d–g). Experiments based on extant caterpillars show that eyespots are costly to small prey because they enhance detectability without providing a protective advantage, but they are beneficial to large prey[47]. Our discovery is consistent with the hypothesis that eyespots are effective deterrents only when both prey and eyespots are large[47].

The present evidence sheds light on the macroevolution of kalligrammatids—a major pollinating insect lineage in the Mesozoic. Kalligrammatid species diversification was potentially promoted by coevolution between pollinating kalligrammatids and their host plants under highly partitioned pollination niches. Traits such as wing eyespots, which likely functioned as a defence in large-sized species, and sexually dimorphic antennae, which were likely used for pre-mating chemical communication, elucidate how kalligrammatids survived in the Mesozoic terrestrial ecosystem under intense competition. However, such elaborate associations between kalligrammatids and their host plants (mostly confined to gymnosperms) could have- been a main factor contributing to the extinction of kalligrammatids, which likely occurred during the late Cretaceous with the decline in gymnosperm diversity.

## Methods

**Materials**. This study refers to 28 new specimens (27 kalligrammatid and 1 *Fiaponeura penghiani*; Lu et al.[24]), including 25 amber specimens and 3 compression fossils. All specimens are deposited in publicly accessible collections. Nineteen specimens (NIGP164470–NIGP164487 and NIGP168262) are housed in the Nanjing Institute of Geology and Palaeontology (NIGP), Chinese Academy of Sciences; two Burmese amber specimens (CAM BA-0010 and CAM BA-0011) are deposited in the Century Amber Museum (CAM) in Shenzhen; two Burmese amber specimens (EMTG BU-002169 and EMTG BU-0022662) are deposited in the Three Gorges Entomological Museum (EMTG) in Chongqing; three Burmese amber specimens (CAU-BA-WN-18001, CAU-BA-XF-18002 and CAU-BA-XF-18003) are deposited in the Entomological Museum, China Agricultural University (CAU), Beijing; and two specimens (STMN45-1614 and STMN45-1624) from Yixian Formation are deposited in the Shandong Tianyu Museum of Nature (Pingyi).

**Imaging**. Specimens were photographed using a Zeiss Stereo Discovery V16 microscope system at the NIGP, Chinese Academy of Sciences. All images were taken by using digitally stacked photomicrographic composites of 40–50 individual focal planes using the image-editing software Helicon Focus 6 [http://www.heliconsoft.com].

**Morphological terminology**. Morphological terminology generally follows Aspöck et al. for the wing venation[17,48] and Aspöck and Aspöck for the genitalia[49]. Abbreviations of wing venation are as follows: A anal vein, Cu cubitus, CuA cubitus anterior, CuP cubitus posterior, MA media anterior, MP media posterior, ORB oblique radial branch, RA radius anterior, RP radius posterior, and ScP subcosta posterior.

**Phylogenetic analysis**. In light of the new taxa and new characters available for further testing the phylogenetic status of the Burmese amber long-proboscid psychopsoids, we reanalysed the phylogenetic relationships among the representative psychopsoids based on Lu et al.[17] by incorporating the presently described new genera and revised character coding into the previous data set of Lu et al.[17] (Supplementary Note 2). Five lacewing genera were selected as the outgroup taxa, including *Nipponeurorthus* (Nevrorthidae), *Dilar* and *Nallachius* (Dilaridae), *Rapisma* (Ithonidae) and *Prohemerobius* (Prohemerobiidae). These families represent relatively basal lineages compared to Psychopsoidea[50,51]. Moreover, Dilaridae was also used to test the familial affinity of Cretanallachiinae that was previously placed in Dilaridae[26]. Specimens of all extant taxa and all Burmese amber psychopsoids were examined by the authors, and the character states of the compression fossil taxa belonging to Aetheogrammatidae, Kalligrammatidae and

Osmylopsychopidae were determined based on reliable and detailed illustrations in the literature[22,52,53].

The morphological characters used in the phylogenetic analysis comprised 33 adult characters (Supplementary Note 3). Unknown characters were coded as '?'. As the presently described new taxa, e.g. *Burmogramma* gen. nov., provide strong evidence to support the kalligrammatid affinity of these Burmese amber psychopsoids with long-proboscid mouthparts, bipectinate male antennae and specialized female tergum 9, the missing data of the antennae and genitalia in most kalligrammatids described based on compression fossils may negatively affect the phylogenetic inference of the relationships within Kalligrammatidae. Therefore, in addition to the primary data set (matrix 1), we made another matrix (matrix 2) with some estimated codes of antenna and genitalia for those kalligrammatid genera missing these characters. Thereinto, as the sexually dimorphic antennae (bipectinate in males) and the strongly expanded female tergum 9 are not found in any kalligrammatids except for the Burmese amber species (see Yang et al.[22]), the relevant characters 26–28 and 31–32 were coded as '0' for *Sophogramma*, *Meioneurites*, *Kallihemerobius*, *Kalligramma* and *Oregramma*. The data matrices are given in Supplementary Data 1 and 2.

We analysed 22 taxa and 33 characters through parsimony analyses performed using WinClada ver. 1.00.08[54]. The heuristic search was used with maximum trees to keep setting to 10,000 and the number of replication setting to 100. To obtain support index for each node, we calculated the bootstrap branch support values using NONA ver. 2.0[55]. All characters were treated as non-additive, unordered and equally weighted.

**Nomenclatural acts.** This published work and the nomenclatural acts it contains have been registered in ZooBank, the proposed online registration system for the International Code of Zoological Nomenclature (ICZN). The ZooBank LSIDs (Life Science Identifiers) can be resolved and the associated information viewed through any standard web browser by appending the LSID to the prefix 'http://zoobank.org/'. The LSIDs for this publication are: urn:lsid:zoobank.org:pub:99146319-D43B-454F-B081-538727680F2A.

## Data availability

The authors declare that the data supporting the findings of this study are available within the paper and its supplementary information files. Higher-resolution versions of the figures have been deposited in the figshare database (https://doi.org/10.6084/m9.figshare.6469385) and can be obtained upon request from the corresponding authors.

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

## Acknowledgements

We are grateful to M.S. Engel, V.N. Makarkin, H.C. Zhang and G.L. Shi for helpful discussions; X. Jia, F.Y. Xia, Y.R. Huang and N. Wang for providing specimens; H. Jiang and X.J. Lei for great assistance with fossil preparation; and D.H. Yang for reconstructions. This research was supported by the National Natural Science Foundation of China (41572010, 31672322, 41622201, 41688103, 31322051) and the Strategic Priority Research Program (B) of the Chinese Academy of Sciences (XDB26000000).

## Author contributions

X.Liu and B.W. designed the project. Q.L., X.Lu, Q.Z., X.Liu and B.W. prepared figures and wrote the manuscript. Q.L., X.Lu, X.Liu and B.W. performed the comparative and analytical work. Q.L., X.Lu, J.C., X.Z. and W.Z. collected data and contributed to the discussion.

## Additional information

**Competing interests:** The authors declare no competing interests.

