## [Peer Review File · Nature Communications]

Reviewers' comments:

Reviewer #1 (Remarks to the Author):

Review of the manuscript "Niche partitioning 1 and chemical communication in Mesozoic pollinating lacewings" by Qing Liu et al. for Nature Communications

Enrique Peñalver

This is an excellent research and I think it is very suitable for its publication in Nature Communications journal. This manuscript presents new impressive, relevant kalligrammatid specimens in amber and compression rocks. The main topics of this manuscript (the earliest pollination niche partitioning and chemical communication) are clearly relevant, but the secondary one as well (eyespot as defense mechanism). The amount of new data and detailed descriptions are impressive, mainly if we consider the Supplementary Information. The manuscript is clear and well-structured in general, despite important problems between the taxonomical data in the main text and the Supplementary Information. Methodology, including the phylogenetic analysis, is correct, but see a comment below about the type repositories. The design of this research seems suitable. In consequence, the manuscript contains important discussions and conclusions. Figures and tables are impressive and very suitable for this manuscript. References are the most relevant for these topics.

I include here a few minor edits, and also some comments to the authors that I consider of importance.

Minor edits:

Line 68: "sediments" changes to "compression rocks". Note that we have not sediments today in the outcrops, because the sediments occurred in these localities during sedimentation and then changed to rocks due to diagenesis (including with lightly compaction and absence of cementation).

Lines 71-73: the sentence is confusing, because the bipectinate male antennae and wing eyespots were not characters acting as adaptive traits for kalligrammatids in plant-pollinating interactions as it indicates.

Line 71 (and 291): use "eye spots" instead of "eyespot" as along the manuscript

Line 77: to delete "etymology" because also is provided in the main text but seems indicate that the etymologies are only present in the Supplementary Information.

Line 91: "Diagnosis" changes to "Diagnosis of the genus (and the types species)" or similar

Line 104: "tremendous" changes to "great"

Lines 106-118: it is important to indicate here that the genus *Burmopsyrops* and its type

species are re-diagnosed in the Supplementary Information.

Line 117: "tremendous" changes to "great"

Line 125: "Diagnosis" changes to "Diagnosis of the genus (and the types species)" or similar

Lines 139-15: it is important to indicate here that the genus *Oligopsychopsis* and its type species are re-diagnosed in the Supplementary Information.

Lines 217-218: better if the sentence changes to "Such features suggest indicate that these palps were probably used to probe for nectar or pollen^{33,34}"

Lines 274-276: better if the reference calls 22-45 are located in "predaceous arthropods" and the call 46 isolated in "vertebrates". In respect to the reference 46, it refers to feathered dinosaurs (instead of lizards, true birds and mammals). Maybe reference 46 discuss about these other vertebrates as showing new predaceous evolved during the Late Cretaceous and in this case that reference call is correct, but, please, revise this detail.

Line 276-277: "Considering the multitude of contemporaneous predators, wing eyespots are an important defensive mechanism in kalligrammatids": the content of this sentence is not suitable (the indication "multitude of contemporaneous predators" is a hollow expression and its link with the importance of the "eyespots defensive mechanism" as well). In any case, "are" changes to "were".

Line 308: it is not clear the correct use of the word "respectively" in this sentence. Apparently it refers to the two different numbers of the specimens, but the authors must be more explicit.

Lines 320-323: Do the journal guidelines allow this treatment of the pictures?

Line 408: "Penñilver" changes to "Peñalver"

Line 410: one minor question, Are the authors X.M. Lu, W.W. Zhang and X.Y. Liu in this reference some of the authors of the present manuscript as I think? In that case, Why they used two initials in the past paper but only one in the present?

Line 445: the word "Insect" in the title must be in lowercase

Line 534: "antennae" changes to "antenna"

Line 537-538: note that the abbreviations are not exactly in alphabetical order

Line 544: "labandeira" changes to "labandeirai". Please, revise all the taxonomical names used in the entire manuscript to be sure that spelling is correct. This circumstance is especially important in one paper in which new taxa is described and named.

Line 549-550: note that the abbreviations are not exactly in alphabetical order. Please, note that I did not revise that circumstance in the figure captions of the Supplementary Information

Comments:

1) Authors must to explain, for example in the Supplementary Information, the reasons to link some males and females into the same species.

2) Figure 5: in my opinion the two illustrations in the figure 5 are artistic illustrations, but not paleoecological reconstructions. I mean, which of the kalligrammatids illustrated ones correspond to the taxa studied? They show not details. What new morphological characters do they show? What about the data used to reconstruct to plants in these illustrations?, etc. In my opinion these illustrations are suitable in other context but not in this technical research.

3) In the section "Materials" (Methods) is not absolutely clear that the four holotypes will be permanently housed in a research institution with strong guaranties to curator them for future reviews (type collection, accessibility...) It is clear that the Nanjing Institute of Geology and Palaeontology (NIGP) is a perfect institution for this purpose (according to the indications of the International Code of Zoological Nomenclature) but it is not clear for the other institutions listed, which seem not strictly museums with the suitable guaranties (at least it was my impression, maybe incorrect, after looking for information of them in the net). This comment concerts mainly to the more important type specimens... the holotypes. Apparently three of the four are housed in the NIGP, but is not completely clear in the text if maybe they will eventually be deposited in other "museum", as indicated for the main part of the paratypes. The holotype specimen CAM BA-0010 is "currently housed in the Entomological Museum, China Agricultural University (CAU), Beijing" but maybe it "will eventually be deposited respectively in the Collection of Xiao Jia in the Century Amber Museum (CAM) in Shenzhen, and the Three Gorges Entomological Museum in Chongqing" and is not clear what of the two collections could be the final repository, apparently the Century Amber Museum; is that institution a suitable repository for holotypes? Please, include a sentence to clearly indicate the final repository of the four holotypes in a clearly suitable institution.

4) This manuscript contains some repeated taxonomical sections in the main text and the Supplementary information. Authors include the diagnoses of the new taxa (the most sensitive and important sections in a taxonomical paper with description of new taxa) in both the main text and the Supplementary Information. Due to evident reasons both repeated texts must be identical in all the details (exact copies), but it is not the case of this manuscript. This is a very important error in this manuscript, but very easy to correct. For example, the diagnosis of the genus *Oligopsychopsis* -and its type species- contains 8 text lines in the main text, but only two in its counterpart in Supplementary Information. This circumstance or error must be perfectly corrected to avoid future serious mistakes and taxonomical problems for the taxonomists

5) In general, it is important to contrast the entire data in the main manuscript and the Supplementary Information in order to avoid contradictions or inconsistencies.

Reviewer #2 (Remarks to the Author):

An interesting paper with some beautiful specimens of Kalligrammatidae. It has a topic that should be of interest to a wide audience.

Below are some comments that should be considered:

Throughout the main manuscript and supplementary data: the naming of the geological epochs should follow the chart of the International Commission on Stratigraphy (ICS) therefore Late Cretaceous should either be late (with a small l) or Upper Cretaceous to conform to the ICS, the same with Early Cretaceous, either early or Lower Cretaceous, Late Jurassic, should be late or Upper Jurassic.

Systematic Palaeontology – Taxonomy seems accurate and valid

B. labandeirai diagnosis: Are there any further diagnostic wing venation characters?

Nanogramma: This name is preoccupied by a species of Caloneurodea (Insecta: Pterygota: Panorthoptera): *Nanogramma Béthoux et al. 2014*, Type species *Nanogramma gandi Béthoux et al. 2014*.

Reference: O. Béthoux, A. Nel, and J. Lapeyrie. 2004. The extinct order Caloneurodea (Insecta: Pterygota: Panorthoptera): Wing venation, systematic and phylogenetic relationships. *Annales Zoologici* 54:289-318.

Therefore, it should be renamed.

Phylogenetic analysis seems reasonable

Discussion:

Mouthparts and pollination niche partitioning

The majority of this section appears to be a description of the mouthparts with little about the niche partitioning.

Last paragraph: Is there anything more specific you can say about the niche partitioning, as the title and abstract of the paper suggest that this is a big theme of the paper, but this is only vaguely referred to in this last paragraph (lines 223-240)? Are there records of any of the plants, with diverse tube lengths, from the deposit, which could possibly have been

pollinated by the kalligrammatids? You give ranges of size for the proboscis lengths, but how significantly different in length do the proboscis have to be to suggest a generalist feeder or an association to a specific plant/niche? Can the taxa be placed in specific niches?

Chemical communication and defence mechanisms

The rarity of ramified antennae (lines 249-252) in Mesozoic insects – is this a true rarity or a result of preservation, especially with specimens preserved in rock?

What is the relevance of figure 5 to the sentence (lines 265-269) about long-distance mate searching behaviour?

Line 275 – would lacewing larvae be predators of kalligrammatids? Wouldn't the listing of a more expected predator be better here, e.g. dragonflies?

Line 280 – 283: Is the mention of eyespots on caterpillars relevant to eyespots on a flying insect? They both have a very different mode of life, and therefore one would expect them to have different predator threats and pressures.

Figures:

All seem fine and sufficiently labelled

References all major expected references present

Supplementary data

Systematic Palaeontology section

Throughout - It would be useful to reference the original publications that the diagnoses are revised from.

The diagnosis of Cretanallachiinae is very long, can this be reduced? Are all listed characters' diagnostic?

With the majority of taxa described, it is hard to see some of the characters described in the figures, due to the small size of the photographs.

Page 16. Remarks

You say "...it should be transferred to Oligopsychoptis based on the presence of most generic diagnostic characters of Oligopsychoptis in this species" It would be useful if you would list these characters.

Oligopsychoptis penniformis

Revised diagnosis

Are there any better characters for the diagnosis of this species than just the wing length and colour pattern? E.g. taken from photographs of the original paper (Chang et al. 2018), or is a re-description necessary?

List of characters

It would be interesting for some commentary on the characters, for example, characters with ranges: Characters 2, 8, 9. E.g. Character 8 – why is (0) 1, (1) 0 and what is the significance of the number of crossveins being less than or more than 20? Why 20?

Figures

On the whole the images are good and drawings appear accurate, and well labelled.

However, some of the images are too small to pick out details from the descriptions, e.g. wing venation in *B. engeli* (figs 1a, c, e) *B. labandeirai* (3a), *O. groehni* (7b), *O. grandis* (8 a, b). There are also no scale bars for some figures (1 a, d, f, h), (2 d, e), (3 f), (5 c), (6 b, d, g), (7, b, d, I, j, n) if they are same size as the accompanying image this should be mentioned in legend, as in the current legend it suggests that they should all have scale bars.

[1] Reviewer #1:

This is an excellent research and I think it is very suitable for its publication in Nature Communications journal. This manuscript presents new impressive, relevant kalligrammatid specimens in amber and compression rocks. The main topics of this manuscript (the earliest pollination niche partitioning and chemical communication) are clearly relevant, but the secondary one as well (eyespot as defense mechanism). The amount of new data and detailed descriptions are impressive, mainly if we consider the Supplementary Information. The manuscript is clear and well-structured in general, despite important problems between the taxonomical data in the main text and the Supplementary Information. Methodology, including the phylogenetic analysis, is correct, but see a comment below about the type repositories. The design of this research seems suitable. In consequence, the manuscript contains important discussions and conclusions. Figures and tables are impressive and very suitable for this manuscript. References are the most relevant for these topics.

Reply: Thanks!

I include here a few minor edits, and also some comments to the authors that I consider of importance.

Minor edits:

[2] Line 68: “sediments” changes to “compression rocks”. Note that we have not sediments today in the outcrops, because the sediments occurred in these localities during sedimentation and then changed to rocks due to diagenesis (including with lightly compaction and absence of cementation).

Reply: Corrected.

[3] Lines 71-73: the sentence is confusing, because the bipectinate male antennae and wing eyespots were not characters acting as adaptive traits for kalligrammatids in plant-pollinating interactions as it indicates.

Reply: Thanks. We have revised this sentence. Please see lines 71-72. “Thus, these characters provide new insight into the niche partitioning, chemical communication, and defence mechanisms of these pollinating insects.”

[4] Line 71 (and 291): use “eye spots” instead of “eyespot” as along the manuscript

Reply: Thanks. We have replaced “eye spots” with “eyespot”.

[5] Line 77: to delete “etymology” because also is provided in the main text but seems indicate that the etymologies are only present in the Supplementary Information.

Reply: Corrected. We retained the etymology parts in the main text and deleted them from the Supplementary Information.

[6] Line 91: “Diagnosis” changes to “Diagnosis of the genus (and the types species)” or similar

Reply: Corrected.

[7] Line 104: “tremendous” changes to “great”

Reply: Corrected.

[8] Lines 106-118: it is important to indicate here that the genus *Burmopsyrops* and its type species are re-diagnosed in the Supplementary Information.

Reply: Done. We have added a “Remarks” part. Please see lines 120-122.

[9] Line 117: “tremendous” changes to “great”

Reply: Corrected.

[10] Line 125: “Diagnosis” changes to “Diagnosis of the genus (and the types species)” or similar

Reply: Corrected.

[11] Lines 139-15: it is important to indicate here that the genus *Oligopsyrops* and its type species are re-diagnosed in the Supplementary Information.

Reply: Done. We have added a “Remarks” part. Please see lines 161-163.

[12] Lines 217-218: better if the sentence changes to “Such features suggest indicate that these palps were probably used to probe for nectar or pollen^{33,34}”

Reply: Done. We replaced “indicate” with “suggest”.

[13] Lines 274-276: better if the reference calls 22-45 are located in “predaceous arthropods” and the call 46 isolated in “vertebrates”. In respect to the reference 46, it refers to feathered dinosaurs (instead of lizards, true birds and mammals). Maybe reference 46 discuss about these other vertebrates as showing new predaceous evolved during the Late Cretaceous and in this case that reference call is correct, but, please, revise this detail.

Reply: Corrected.

[14] Line 276-277: “Considering the multitude of contemporaneous predators, wing eyespots are an important defensive mechanism in kalligrammatids”: the content of this sentence is not suitable (the indication “multitude of contemporaneous predators” is a hollow expression and its link with the importance of the “eyespots defensive mechanism” as well). In any case, “are” changes to “were”.

Reply: Thanks. We have revised this sentence. Please see lines 284-285. “Wing eyespots were an important defensive mechanism in kalligrammatids.”

[15] Line 308: it is not clear the correct use of the word “respectively” in this sentence. Apparently it refers to the two different numbers of the specimens, but the authors must be more explicit.

Reply: Corrected. Please see lines 310-320.

[16]Lines 320-323: Do the journal guidelines allow this treatment of the pictures?

Reply: We digitally stacked the slides to obtain a better 3D image, and did not change the original color or structure. The final stacked image represents the original data and conforms to the journal and community standards.

[17]Line 408: “Penñilver” changes to “Peñalver”

Reply: Corrected.

[18]Line 410: one minor question, Are the authors X.M. Lu, W.W. Zhang and X.Y. Liu in this reference some of the authors of the present manuscript as I think? In that case, Why they used two initials in the past paper but only one in the present?

Reply: Corrected. We have changed three initials into two in the revised version of both main text and Supplementary Information.

[19]Line 445: the word “Insect” in the title must be in lowercase

Reply: Corrected.

[20]Line 534: “antennae” changes to “antenna”

Reply: Corrected.

[21]Line 537-538: note that the abbreviations are not exactly in alphabetical order

Reply: Corrected.

[22]Line 544: “labandeira” changes to “labandeirai”. Please, revise all the taxonomical names used in the entire manuscript to be sure that spelling is correct. This circumstance is especially important in one paper in which new taxa is described and named.

Reply: Corrected.

[23]Line 549-550: note that the abbreviations are not exactly in alphabetical order. Please, note that I did not revise that circumstance in the figure captions of the Supplementary Information

Reply: Thanks. Corrected.

Comments:

[24] Authors must to explain, for example in the Supplementary Information, the reasons to link some males and females into the same species.

Reply: Thanks. For the species herein studied, it is not difficult to find evidence to link the males and females into same species. We have added a detailed explanation on how to associate the conspecific males and females in the Supplementary Information. Please see lines 105-114 in the Supplementary Information. “Association between conspecific males and females was primarily based on the similarity of body size, wing venation, and wing marking pattern.

The combination of characters of body-size, wing venations and wing marking patterns is in general stable among conspecific males and females but differs among different species. For example, *B. liui* and *F. penghiani* have very characterized wing markings and venations, which can facilitate the association of males and females. Similarly, *C. magnificus* is the smallest species of Cretanallachiinae known so far and has no wing marking. So, the newly found female of this species was linked to the male by its very small body-size (smaller than females of any other cretanallachiine species) and the immaculate wings.”

[25] Figure 5: in my opinion the two illustrations in the figure 5 are artistic illustrations, but not paleoecological reconstructions. I mean, which of the kalligrammatids illustrated ones correspond to the taxa studied? They show not details. What new morphological characters do they show? What about the data used to reconstruct to plants in these illustrations?, etc. In my opinion these illustrations are suitable in other context but not in this technical research.

Reply: We fully agreed with the reviewer and have deleted the figure 5.

[26] In the section “Materials” (Methods) is not absolutely clear that the four holotypes will be permanently housed in a research institution with strong guaranties to curator them for future reviews (type collection, accessibility...) It is clear that the Nanjing Institute of Geology and Palaeontology (NIGP) is a perfect institution for this purpose (according to the indications of the International Code of Zoological Nomenclature) but it is not clear for the other institutions listed, which seem not strictly museums with the suitable guaranties (at least it was my impression, maybe incorrect, after looking for information of them in the net). This comment concerns mainly to the more important type specimens... the holotypes. Apparently three of the four are housed in the NIGP, but is not completely clear in the text if maybe they will eventually be deposited in other “museum”, as indicated for the main part of the paratypes. The holotype specimen CAM BA-0010 is “currently housed in the Entomological Museum, China Agricultural University (CAU), Beijing” but maybe it “will eventually be deposited respectively in the Collection of Xiao Jia in the Century Amber Museum (CAM) in Shenzhen, and the Three Gorges Entomological Museum in Chongqing” and is not clear what of the two collections could be the final repository, apparently the Century Amber Museum; is that institution a suitable repository for holotypes? Please, include a sentence to clearly indicate the final repository of the four holotypes in a clearly suitable institution.

Reply: Both the Century Amber Museum (CAM) and Three Gorges Entomological Museum are permanent and publicly accessible museums, and some type specimens of other Burmese amber insects have already been deposited in these two museums (e.g., type specimens in Liu et al., 2016, Current Biology; Bai et al., 2018, Current Biology). We have revised these sentences to clearly show the deposition locations. Please see lines 310-315. “two Burmese amber specimens (CAM BA-0010 and CAM BA-0011) are deposited in the

Collection of Xiao Jia in the Century Amber Museum (CAM) in Shenzhen; two Burmese specimens (EMTG BU-002169, and EMTG BU-0022662) are deposited in the Three Gorges Entomological Museum (EMTG) in Chongqing”. As the reviewer may know, China is becoming the biggest market for trade of Burmese amber. More and more scientifically important ambers are found to be housed in personal collections. There are very few national institutions, such as NIGP, that can afford to buy these very expensive ambers. Fortunately, some Chinese amber collectors (e.g. Xiao Jia and Weiwei Zhang) have built publicly accessible museums that are open to all amber researchers for checking their amber specimens. Although these museums are new and may at this moment lack information on internet, they are absolutely suitable for deposition and correspond to all requirements of type collection for future reviews, and they will become better in near future. Their amber collections will be at any rate of very important value for the scientific research.

[27] This manuscript contains some repeated taxonomical sections in the main text and the Supplementary information. Authors include the diagnoses of the new taxa (the most sensitive and important sections in a taxonomical paper with description of new taxa) in both the main text and the Supplementary Information. Due to evident reasons both repeated texts must be identical in all the details (exact copies), but it is not the case of this manuscript. This is a very important error in this manuscript, but very easy to correct. For example, the diagnosis of the genus *Oligopsychopsis* -and its type species- contains 8 text lines in the main text, but only two in its counterpart in Supplementary Information. This circumstance or error must be perfectly corrected to avoid future serious mistakes and taxonomical problems for the taxonomists

Reply: Thanks. We have deleted the repeated texts (including diagnosis, type materials, and etymology) in the Supplementary Information.

[28] In general, it is important to contrast the entire data in the main manuscript and the Supplementary Information in order to avoid contradictions or inconsistencies.

Reply: Thanks. We carefully checked the data in the main manuscript and Supplementary Information. We have deleted the repeated texts (including diagnosis, type materials, and etymology) in the Supplementary Information.

Reviewer #2 (Remarks to the Author):

[29] An interesting paper with some beautiful specimens of Kalligrammatidae. It has a topic that should be of interest to a wide audience.

Reply: Thanks!

Below are some comments that should be considered:

[30] Throughout the main manuscript and supplementary data: the naming of the geological epochs should follow the chart of the International Commission on Stratigraphy (ICS) therefore Late Cretaceous should either be late (with a small l)

or Upper Cretaceous to conform to the ICS, the same with Early Cretaceous, either early or Lower Cretaceous, Late Jurassic, should be late or Upper Jurassic.
Reply: Corrected.

[31] Systematic Palaeontology – Taxonomy seems accurate and valid

Reply: Thanks.

[32] *B. labandeirai* diagnosis: Are there any further diagnostic wing venation characters?

Reply: Based on this specimen (the holotype), we could not find any further diagnostic wing venation character of this new species. In most cases, the wing venation is not changed or barely changed between different lacewing species in a same genus. The diagnostic characters to define congeneric species mainly consist of body-size, marking patterns on body and wings, and genitalia.

[33] *Nanogramma*: This name is preoccupied by a species of *Caloneurodea* (Insecta:

Pterygota: Panorthoptera): *Nanogramma* Béthoux et al. 2014, Type species *Nanogramma gandi* Béthoux et al. 2014.

Reference: O. Béthoux, A. Nel, and J. Lapeyrie. 2004. The extinct order *Caloneurodea* (Insecta: Pterygota: Panorthoptera): Wing venation, systematic and phylogenetic relationships. *Annales Zoologici* 54:289-318.

Therefore, it should be renamed.

Reply: Thanks a lot! We have replaced “*Nanogramma*” with “*Cretogramma*” throughout the paper.

Discussion:

[34] Mouthparts and pollination niche partitioning

The majority of this section appears to be a description of the mouthparts with little about the niche partitioning.

Last paragraph: Is there anything more specific you can say about the niche partitioning, as the title and abstract of the paper suggest that this is a big theme of the paper, but this is only vaguely referred to in this last paragraph (lines 223-240)? Are there records of any of the plants, with diverse tube lengths, from the deposit, which could possibly have been pollinated by the kalligrammatids? You give ranges of size for the proboscis lengths, but how significantly different in length do the proboscis have to be to suggest a generalist feeder or an association to a specific plant/niche? Can the taxa be placed in specific niches?

Reply: Thanks. The suggestion of the reviewer is illuminating! We fully agreed with the reviewer that the plants are very important for studying niche partitioning. However, little is known about the morphology of Mesozoic floral tubes because floral tubes are hardly preserved. For example, we have not found the floral tubes from Burmese amber so far. Ren et al. (2009, *Science*) provided an overview of Mesozoic floral tubes including only 5 plants. There is almost no other relevant literature. So we cannot give a detailed discussion about specific

plants/niches for the moment. Nevertheless, our present finding could represent a prediction of diverse tube lengths of the Burmese amber plants. Future works on the Burmese amber plants may provide evidence to test our hypothesis.

[35] Chemical communication and defence mechanisms

The rarity of ramified antennae (lines 249-252) in Mesozoic insects – is this a true rarity or a result of preservation, especially with specimens preserved in rock?

Reply: The rarity of ramified antennae is a true rarity, because ramified antennae have no negative effects on the preservation.

[36] What is the relevance of figure 5 to the sentence (lines 265-269) about long-distance mate searching behaviour?

Reply: Thanks. We have deleted the figure 5.

[37] Line 275 – would lacewing larvae be predators of kalligrammatids? Wouldn't the listing of a more expected predator be better here, e.g. dragonflies?

Reply: Corrected. We have replaced “lacewing larvae” with “dragonflies”.

[38] Line 280 – 283: Is the mention of eyespots on caterpillars relevant to eyespots on a flying insect? They both have a very different mode of life, and therefore one would expect them to have different predator threats and pressures.

Reply: Thanks. Hossie et al. (2015, PNAS) did not test their hypothesis about the flying insects. Caterpillars and kalligrammatids have different predator threats and pressures, but their eyespots most likely have the same function because the eyespots on wings probably function for alerting when the insect is not flying. Hossie's hypothesis is well compatible with our observations of fossil kalligrammatids, and therefore, is probably widespread in both nonflying and flying insects.

[39] Supplementary data

Systematic Palaeontology section

Throughout - It would be useful to reference the original publications that the diagnoses are revised from.

Reply: Thanks. We have added the original references in the revised diagnoses.

[40] The diagnosis of Cretanallachiinae is very long, can this be reduced? Are all listed characters' diagnostic?

Reply: We have deleted “distinctly widened distad” in this section. But for the other characters, they are really useful to define the subfamily, so we consider retaining them.

[41] With the majority of taxa described, it is hard to see some of the characters described in the figures, due to the small size of the photographs.

Reply: We have uploaded the high-res images of these taxa to the figshare database. Please see lines 379-381. “Higher resolution versions of the figures have been deposited in the figshare database (DOI: 10.6084/m9.figshare.6469385; <https://figshare.com/s/0418a65837654d17fffa>) and can be obtained upon request from the corresponding authors.”

[42]Page 16. Remarks

You say “...it should be transferred to *Oligopsychoptis* based on the presence of most generic diagnostic characters of *Oligopsychoptis* in this species” It would be useful if you would list these characters.

Reply: Corrected. Please see Supplementary Information lines 478-480. “..., such as the presence of forewing median nygma, the deeply branched forewing MA, and the presence of sigmoid stem of hind wing MA.”

[43]*Oligopsychoptis penniformis*

Revised diagnosis

Are there any better characters for the diagnosis of this species than just the wing length and colour pattern? E.g. taken from photographs of the original paper (Chang et al. 2018), or is a re-description necessary?

Reply: Based on our re-examination of the holotype, we could not find any further diagnostic character of this species. In most cases, the wing venation is not changed or barely changed between different lacewing species in a same genus. At the species level, the diagnostic characters to define congeneric species of Neuroptera mainly consist of body-size, marking patterns on body and wings, and genitalia, while the venations are usually useless to distinguish congeneric species.

[44]List of characters

It would be interesting for some commentary on the characters, for example, characters with ranges: Characters 2, 8, 9. E.g. Character 8 – why is (0) 1, (1) 0 and what is the significance of the number of crossveins being less than or more than 20? Why 20?

Reply: We appreciate the suggestion. We have added comments on the characters 2, 6, 7, 8, 9, 21. For other characters it is easy to understand the difference between the states and the polarization.

[45]Figures

On the whole the images are good and drawings appear accurate, and well labelled.

However, some of the images are too small to pick out details from the descriptions, e.g. wing venation in *B. engeli* (figs 1a, c, e) *B. labandeirai* (3a), *O. groehni* (7b), *O. grandis* (8 a, b). There are also no scale bars for some figures (1 a, d, f, h), (2 d, e), (3 f), (5 c), (6 b, d, g), (7, b, d, I, j, n) if they are same size as the accompanying image this should be mentioned in legend, as in the current

legend it suggests that they should all have scale bars.

Reply: Thanks. We have revised the figure legends.

REVIEWERS' COMMENTS:

Reviewer #1 (Remarks to the Author):

I reviewed carefully the answers to all the referees' questions and the changes in the two parts of the manuscript and, in my opinion, all the answers and changes are correct and have been done in a constructive manner.

Enrique Peñalver

Reviewer #2 (Remarks to the Author):

All comments seem to have been addressed in the manuscript and supplementary info.

A couple of small changes needed are:

Pg2 Line 67: Early needs to be "early" or "Lower"

Pg 5 Line 140: the etymology definition needs changing – you still refer to the small body size which was with respect to the etymology of the previous name "Nannogramma".

Reviewer 2:

[1] Pg2 Line 67: Early needs to be “early” or “Lower”

Reply: Thanks. Corrected.

[2] Pg 5 Line 140: the etymology definition needs changing – you still refer to the small body size which was with respect to the etymology of the previous name “Nannogramma”.

Reply: Corrected.